# Cardiac Ischaemia–Reperfusion Injury: Pathophysiology, Therapeutic Targets and Future Interventions

**DOI:** 10.3390/biomedicines13092084

**Published:** 2025-08-27

**Authors:** Lujain Alsadder, Abdulaziz Hamadah

**Affiliations:** Faculty of Medicine and Dentistry, Queen Mary University of London, London E1 4NS, UK; a.s.a.hamadah@smd20.qmul.ac.uk

**Keywords:** ischaemia–reperfusion injury, cardiovascular disease, pathophysiology

## Abstract

Advancements in the medical field, particularly in cardiovascular diseases, have significantly improved the diagnosis, management, and prevention of life-threatening presentations and comorbidities. Despite this progress, cardiovascular diseases continue to place a substantial burden on healthcare systems, contributing to nearly 32% of all global deaths according to the World Health Organisation. A predominant complication arising from the treatment of cardiovascular diseases is cardiac ischaemia–reperfusion (I/R) injury, which occurs when blood supply is restored to the myocardium following a period of ischaemia, paradoxically resulting in further tissue damage. There are multiple factors involved in complex pathophysiology and complicated clinical outcomes. Although various therapeutic strategies have been explored to mitigate this injury, an optimal solution has yet to be identified. Therapeutic approaches such as pharmacological interventions and molecular therapy have shown promising prospects in this field. Ongoing research aims to address this unresolved issue, which continues to pose significant challenges for both patients and healthcare professionals. This review aims to explore the multitude of underlying mechanisms of ischaemia–reperfusion injury, and identify current knowledge gaps and new emerging therapeutic interventions.

## 1. Introduction

Cardiac ischaemia, a huge driver of ischaemic heart disease (IHD), arises when myocardial oxygen supply is restricted due to inadequate blood flow, most commonly due to a coronary artery obstruction from atherosclerotic plaque [1,2]. Ischaemia leads to a series of biochemical disturbances, including reduced ATP production, intracellular acidosis, and the generation of reactive oxygen species (ROS), which compromise myocardial function and viability [3]. Sustained, severe ischaemia results in myocardial infarction (MI) and subsequent cardiac dysfunction and potentially leads to heart failure [4]. Silent ischaemia, which is common among patients with diabetes or autonomic neuropathy, presents distinctive diagnostic challenges due to the presence of atypical symptoms, further complicating early detection and resulting in profound complications [1].

Cardiac ischaemia has also contributed significantly to global cardiovascular morbidity and mortality numbers. The World Health Organisation [5] recently reported that IHD results in around 9 million deaths annually, accounting for 16% mortality worldwide. Countries with low- and middle-income bear a disproportionate burden, with over 75% of deaths occurring in these regions, reflecting disparities in healthcare access and the rising prevalence of risk factors such as hypertension, hyperlipidaemia, and diabetes [5]. Despite advancements in medical therapies and revascularisation strategies, cardiac ischaemia continues to impose substantial healthcare costs, largely due to the long-term reduction in quality of life for patients [6]. Thus, tackling this significant burden requires comprehensive prevention, early detection, and equitable access to evidence-based therapies [4].

### Significance of Ischaemia–Reperfusion Injury

Investigating ischaemia–reperfusion (I/R) injury is essential for improving the understanding and management of MI, as well as enhancing outcomes following percutaneous coronary intervention (PCI), cardiac surgery and heart transplantation. I/R injury occurs when blood flow is restored to the myocardium following a period of ischaemia, paradoxically causing additional cellular and tissue damage [7]. In the context of MI, reperfusion therapies, such as PCI, are lifesaving; nevertheless, they can trigger I/R injury leading to adverse remodelling and heart failure [8]. This paradox highlights the need for targeted strategies to mitigate I/R injury and improve long-term outcomes for MI patients.

In cardiac surgery, including coronary artery bypass grafting (CABG) and heart transplantation, I/R injury has a significant effect on patient outcomes. During these procedures, the myocardium often experiences temporary global ischaemia due to surgical aortic clamping, followed by reperfusion, which can precipitate arrhythmia, tissue damage, and contractile dysfunction [9]. In transplantation, cold ischaemia, a surgical technique, involves organs cooling after blood supply has been cut off to reduce the organ metabolism and oxygen consumption, but the subsequent reperfusion contributes to graft dysfunction and rejection [10]. Therefore, improved understanding of I/R injury mechanisms will help in the development of cardioprotective interventions and improve procedural success and patient survival rates [11].

## 2. Objectives and Scope of the Review

The primary aim of this review is to explore the mechanisms, clinical implications, and potential therapeutic strategies associated with cardiac I/R injury, with a specific emphasis on its role following acute MI. This review seeks to elucidate the complex cellular and molecular pathways underpinning I/R injury, including oxidative stress, inflammation, and mitochondrial dysfunction. Furthermore, it aims to evaluate emerging cardioprotective strategies, such as ischaemic conditioning techniques (pre- and post-conditioning), pharmacological interventions, and biomolecular therapies, and their translational potential in mitigating myocardial damage.

The scope of this review is confined to experimental and clinical studies published over the past decade to maintain a focus on contemporary developments. It includes a critical assessment of ongoing preclinical models, translational research, and clinical trials to identify existing translational challenges and knowledge gaps, such as the interplay between mitochondrial dysfunction and the multiple forms of cell death, targeting inflammation and oxidative damage, and suggests future research directions.

## 3. Pathophysiology of Ischaemia–Reperfusion Injury

### 3.1. Cellular and Molecular Changes

Ischaemia–reperfusion injury (IRI) occurs in the setting of inadequate blood supply, restricting oxygen and nutrient delivery to the myocardium (ischaemia) with subsequent restoration of blood flow (reperfusion), leading to paradoxical tissue damage [7]. During the ischaemic phase, reduced oxygen delivery disrupts oxidative phosphorylation in mitochondria, the primary energy source for cardiomyocytes [12]. This shift to anaerobic glycolysis results in significantly reduced ATP production, impairing cellular energy-dependent functions [13]. Synchronously, the breakdown of glycogen and accumulation of glycolytic by-products, such as lactate and hydrogen ions, alter intracellular pH, leading to acidosis [14]. This metabolic shift destabilises cellular homeostasis, contributing to ionic changes, mitochondrial membrane depolarisation and activating early necrotic pathways [15]. The oxygen shortage during ischaemia also inhibits electron transport chain (ETC) activity, causing electron leakage, setting the stage for reactive oxygen species (ROS) overproduction during reperfusion [16].

ATP depletion aggravates cellular dysfunction by impairing critical ion transport mechanisms. With diminished sodium-potassium ATPase activity, intracellular sodium levels rise, driving subsequent calcium entry through the sodium-calcium exchanger [17]. These ionic changes lead to calcium accumulation in the mitochondria, which, if sustained for prolonged ischemic durations, can disrupt mitochondrial function and promote mitochondrial permeability transition pore (mPTP) opening despite intracellular acidosis, as demonstrated in Figure 1 [18]. This can contribute to cellular swelling, rupture, and eventually cell death [19]. Meanwhile, acidification from ischaemic by-products activates acid-sensing ion channels (ASICs), further destabilising intracellular ion gradients and amplifying protease activation [20]. These molecular changes initiate necrosis and apoptosis, while the activation of damage-associated molecular patterns (DAMPs) during reperfusion triggers inflammatory cascades, compounding tissue injury [14].

Cooperatively, these cellular and molecular derangements in ion homeostasis, ATP depletion, and oxygen deprivation during ischaemia lead to widespread cardiomyocyte death and impaired cardiac function. Paradoxically, further injury is seen during treatment due to reperfusion.

### 3.2. The Paradoxical Damage of Reperfusion

During cardiac reperfusion, the abrupt restoration of blood flow leads to a marked increase in intracellular calcium, a surge in mitochondrial ROS production, which opens the mitochondrial permeability transition pore that leads to necrotic, apoptotic and necroptotic myocardial cell death [23,24]. Mitochondria, as primary energy producers, become a significant ROS source when oxygen supply is reinstated, resulting in the initiation of superoxide anions and hydrogen peroxide [16]. This oxidative milieu damages cellular components, including lipids, proteins, and DNA, thereby impairing cardiac function. Additionally, ROS activate signalling pathways that stimulate inflammation and cell death, further contributing to tissue injury [25]. Recent studies have highlighted the pivotal role of oxidative stress in mediating reperfusion injury, highlighting the therapeutic potential of targeting ROS to mitigate cardiac damage [26].

The imbalance between the formation of ROS and the antioxidant defence system during reperfusion leads to oxidative modifications of critical cellular structures [21]. This imbalance disrupts ion homeostasis, predominantly calcium handling, exacerbating myocardial injury. Interventions aimed at enhancing endogenous antioxidant capacity or directly scavenging ROS have shown promising outcomes in experimental models, suggesting potential avenues for clinical translation [27].

Calcium overload during reperfusion paradoxically exacerbates myocardial injury despite restoring oxygen supply. While the re-establishment of blood flow is vital to salvage ischaemic tissue, it also triggers abrupt calcium influx into cardiomyocytes, which were already primed by ionic imbalances during ischaemia [12]. As mentioned previously, the sudden calcium surge leads to mitochondrial calcium overload, inducing the pathological opening of the mPTP and initiating cell death pathways [15]. Furthermore, calcium overload activates calpains, calcium-dependent proteases, which degrade cytoskeletal and membrane proteins, thereby contributing to structural damage and impaired contractility [8]. This paradox highlights the dual nature of reperfusion and emphasises the need for targeted treatments to avoid calcium-mediated destruction.

### 3.3. Inflammatory Response

The long-term pathophysiological changes associated with cardiac I/R are influenced by immune activation, with cytokines, neutrophils, and immune signalling playing fundamental roles. During reperfusion, DAMPs released by injured cardiomyocytes activate pattern recognition receptors, such as Toll-like receptors (TLRs), stimulating inflammatory cascades [28]. Pro-inflammatory cytokines, including tumour necrosis factor-alpha (TNF-α) and interleukin-6 (IL-6), are rapidly upregulated, amplifying the inflammatory response and recruiting neutrophils to the myocardium [29].

Neutrophils, in turn, exacerbate myocardial injury by releasing proteolytic enzymes, ROS, and pro-inflammatory mediators [30], which cause endothelial dysfunction, microvascular obstruction, and additional cardiomyocyte damage. In addition, immune signalling pathways, such as nuclear factor-kappa B (NF-κB) activation, perpetuate inflammation, impairing myocardial repair [31]. Modulating inflammatory responses, including inhibiting neutrophil infiltration and cytokine release, has shown potential in experimental models to moderate cardiac IRI [32].

As cardiac IRI is tightly linked to the pro-inflammatory cascades driven by cytokines, neutrophils, and immune signalling, reperfusion-induced release of DAMPs activates pattern recognition receptors, amplifying cytokine production and neutrophil infiltration [27,33]. This sterile inflammatory response exacerbates myocardial injury by endorsing inflammasome activation, particularly NOD-like receptor protein 3 (NLRP3), which drives the release of interleukin-1β (IL-1β) and interleukin-18 (IL-18) [34]. These cytokines preserve neutrophil-mediated endothelial dysfunction and microvascular obstruction, worsening myocardial necrosis [30]. Additionally, neutrophil extracellular traps (NETs), a network of extracellular DNA, that are formed during sterile inflammation, aggravate tissue damage by amplifying oxidative damage and impairing cardiac repair [35]. This dynamic interaction between sterile inflammation and immune activation emphasises its critical role in the pathophysiology of cardiac IRI.

### 3.4. Apoptosis and Necrosis

Cardiac IRI induces different types of cell death mechanisms, including apoptosis and necrosis, highlighting their significant impact on clinical outcomes. Apoptosis, a programmed cell death pathway, is primarily generated during reperfusion by mitochondrial dysfunction, including the release of cytochrome c and activation of caspases [36]. Concurrently, necrosis, an unregulated method of cell death, arises due to severe ATP depletion and plasma membrane rupture, leading to the release of DAMPs that perpetuate inflammation, as mentioned previously [37]. Both pathways interact synergistically, exacerbating myocardial injury, with apoptosis contributing to delayed cell loss and necrosis, causing immediate structural damage during reperfusion [23].

Furthermore, there is emerging evidence that identifies novel contributors to cardiac IRI, such as necroptosis and ferroptosis, as illustrated in Figure 2, as well as autophagy and cuproptosis [38]. Ferroptosis, a regulated cell death mechanism driven by iron-dependent lipid peroxidation, has been implicated in cardiomyocyte injury during reperfusion [7]. The excessive ROS and impaired antioxidant defences, such as glutathione peroxidase 4 (GPX4), exacerbate ferroptotic damage, which can be targeted with new therapeutic interventions [39]. Necroptosis, on the other hand, a programmed form of necrosis mediated by receptor-interacting protein kinases (RIPK1 and RIPK3), amplifies myocardial damage [24] through membrane rupture and inflammation [40]. These emerging insights provide opportunities to develop innovative therapies by targeting ferroptotic and necroptotic pathways to alleviate myocardial injury in I/R.

Autophagy, a cellular degradation and recycling process, is one of the main factors that play a crucial role in the mechanisms underlying IRI. Autophagy has a dual role in IRI, exerting both protective and detrimental effects depending on its regulation and extent. During ischaemia, autophagy is generally protective, mitigating cellular stress by removing damaged organelles and misfolded proteins, thereby preserving mitochondrial function and energy homeostasis [43]. This adaptive response enhances cell survival by preventing the accumulation of toxic by-products that would otherwise exacerbate injury. However, excessive or dysregulated autophagy during reperfusion can become detrimental, leading to excessive degradation of essential cellular components and promoting autophagic cell death [8]. Overactivation of autophagy, often driven by persistent AMP-activated protein kinase (AMPK) activation and Beclin-1 upregulation, can aggravate myocardial damage [21]. Thus, modulating autophagy to balance its protective and harmful effects presents a promising therapeutic strategy for mitigating IRI while preserving cardiac function.

### 3.5. Oxidative Stress

Oxidative stress is a key molecular mechanism underlying cardiac IRI, primarily driven by excessive ROS generation during reperfusion. The major sources of ROS, as shown in Figure 3, include mitochondrial dysfunction, xanthine oxidase activation, NADPH oxidase in infiltrating neutrophils, and uncoupled nitric oxide synthase [17]. Mitochondrial ROS generation is specifically critical, as impaired ETC function leads to superoxide production, which can be converted into more reactive molecules such as hydrogen peroxide and hydroxyl radicals [16].

ROS-induced oxidative damage targets fundamental cellular components. Lipid peroxidation compromises membrane integrity by altering phospholipid composition, increasing permeability, and generating toxic by-products such as malondialdehyde (MDA) and 4-hydroxynonenal (4-HNE), which further worsen oxidative injury [44]. Proteins are highly susceptible to ROS modifications, particularly thiol oxidation, carbonylation, and nitration, which impair enzymatic function and structural stability, contributing to contractile dysfunction and metabolic derangements [45]. DNA is also a major target, with ROS causing strand breaks and base modifications such as 8-hydroxydeoxyguanosine (8-OHdG), leading to genomic instability and activation of pro-apoptotic pathways [46]. Additionally, oxidative stress activates inflammatory signalling cascades, exacerbating myocardial injury and impairing cardiac repair [8]. Understanding the molecular basis of oxidative stress in IRI highlights the importance of therapeutic strategies targeting ROS generation and antioxidant defences. Novel interventions, including mitochondrial-targeted antioxidants and enzyme modulators, offer promising avenues to mitigate oxidative injury in IRI [47,48].

### 3.6. Endothelial Dysfunction

Nitric oxide (NO) is a fundamental regulator of vascular homeostasis, synthesised by endothelial nitric oxide synthase (eNOS) in endothelial cells and has an essential role in the pathophysiological mechanisms underlying IRI. Oxidative stress leads to a reduction in NO bioavailability, resulting in impaired vasodilation and increased vascular permeability [49]. Furthermore, the reduced NO levels favour leukocyte adhesion and infiltration, amplifying the inflammatory response and subsequent tissue damage [29]. Additionally, endothelial dysfunction leads to increased vascular permeability, permitting plasma proteins and fluids to extravasate into the interstitial space, resulting in tissue oedema [21]. This oedema increases interstitial pressure, compresses capillaries, and impairs microvascular perfusion, exacerbating myocardial ischemia and hindering oxygen delivery to cardiomyocytes, similar to the response observed in microvascular obstruction [50]. The compromised endothelial barrier function is associated with the degradation of the glycocalyx, a protective layer on the endothelial surface, which further enhances permeability and promotes leukocyte adhesion and infiltration [15]. The infiltrating leukocytes release proteolytic enzymes and ROS, amplifying tissue injury and inflammation. Therapeutic strategies aimed at preserving endothelial integrity and reducing vascular permeability have potential promise in mitigating myocardial damage in IRI [51].

Furthermore, endothelial-to-mesenchymal transition (EndMT) is another example of pathological processes contributing to endothelial dysfunction in cardiac IRI. Inflammatory cytokines and oxidative stress activate signalling pathways during IRI, notably transforming growth factor-beta (TGF-β), which induces endothelial cells to lose their typical markers and acquire mesenchymal-like properties [27]. This conversion results in increased fibroblast-like activity, excessive extracellular matrix deposition, and myocardial fibrosis, ultimately impairing cardiac function [12]. Additionally, EndMT compromises endothelial barrier integrity, worsening vascular permeability and inflammation in the reperfused myocardium [39]. Hence, understanding the crucial role of EndMT in IRI provides new therapeutic insights into improving endothelial resilience and reducing myocardial damage.

### 3.7. Mitochondrial Dysfunction

The mPTP plays a critical role in mitochondrial dysfunction during cardiac IRI. Located in the inner mitochondrial membrane, the mPTP remains closed under most complex physiological conditions, preserving mitochondrial membrane potential and ATP production [41]. However, during reperfusion, factors such as calcium overload, oxidative stress, and ATP depletion trigger mPTP opening [29]. This leads to the dissipation of the mitochondrial membrane potential, uncoupling of oxidative phosphorylation, and a consequent drop in ATP synthesis [15]. Additionally, mPTP opening facilitates the release of pro-apoptotic factors, such as cytochrome c, which activate caspase-dependent apoptotic pathways [31]. The prolonged opening of mPTP results in irreversible mitochondrial swelling, membrane rupture, and finally, cell death via necrosis or apoptosis of the cardiac tissue. As an insight into the therapeutic approaches for this mechanism, pharmacological agents, such as cyclosporine A, have been investigated for their ability to inhibit mPTP opening and reduce myocardial damage, highlighting the therapeutic relevance in mitigating cardiac IRI [52].

Furthermore, mitochondrial dysfunction has a vital role in the molecular mechanisms underlying cardiac IRI, as mitochondrial dynamics and quality control are critical regulators of mitochondrial function. Mitochondrial dynamics involve continuous cycles of fission and fusion, processes essential for maintaining mitochondrial integrity and adapting to cellular stress [37]. During IRI, excessive mitochondrial fission, often mediated by dynamin-related protein 1 (Drp1), leads to mitochondrial fragmentation, contributing to dysfunction and increased susceptibility to cell death [36].

In parallel, mitochondrial quality control mechanisms, including mitophagy, remove damaged mitochondria to prevent the accumulation of dysfunctional organelles [53]. However, impaired mitophagy during reperfusion can worsen oxidative stress and mitochondrial dysfunction, amplifying myocardial injury [37]. The regulation of mitophagy is tightly controlled by pathways involving PINK1 (PTEN-induced putative kinase 1) and Parkin, which tag depolarised mitochondria for degradation via autophagosomes [54]. Disruption in these signalling pathways during IRI impairs clearance of damaged mitochondria, exacerbating cardiomyocyte injury. Similarly, the fission/fusion balance is modulated by the activity of proteins such as mitofusins 1/2 (Mfn1/2) and optic atrophy 1 (Opa1), which promote fusion, and Drp1, which promotes fission. Abnormal phosphorylation or translocation of Drp1 during reperfusion has been linked to pathological fission and mitochondrial fragmentation [55].

Pharmacological modulation of these pathways—for example, using Drp1 inhibitors like Mdivi-1 or agents enhancing PINK1-Parkin signalling—has shown protective effects in preclinical IRI models by stabilising mitochondrial networks and enhancing mitophagic flux. These findings underscore the potential of targeting mitochondrial quality control as a viable therapeutic strategy in myocardial IRI [56].

Strategies aimed at restoring mitochondrial dynamics or enhancing mitophagy have shown potential in preserving mitochondrial function and reducing cell death, positioning mitochondrial quality control as a promising therapeutic target in IRI [57].

## 4. Experimental Models in I/R Research

### 4.1. Ex Vivo Models

Significant progress has been made in the study of IRI with experimental models, with *ex vivo* models being extensively investigated to advance understanding in this field. Cardiomyocyte cultures and hypoxia-reoxygenation (H/R) assays are extensively used models for studying IRI, providing valuable insights into the cellular and molecular mechanisms underlying myocardial damage [58]. These models offer a controlled environment to explore pathophysiological processes, including oxidative stress, mitochondrial dysfunction, calcium overload, and cell death pathways [59].

Primary cardiomyocytes derived from neonatal or adult hearts, as well as cardiomyocyte-like cell lines (e.g., H9c2 cells), function as *in vitro* platforms to simulate IRI [60]. In these models, hypoxia is provoked by mechanisms like oxygen-glucose deprivation (OGD) or hypoxic chambers, imitating ischaemic conditions by limiting oxygen and nutrient supply [61]. Subsequent reoxygenation restores normal oxygen levels in the cardiomyocyte, simulating reperfusion and activating cellular stress responses, such as ROS generation, apoptosis, and inflammation [26]. Cardiomyocyte cultures have many advantages, including high reproducibility, controlled experimental conditions, and the capability to test pharmacological interventions. H/R assays facilitate mechanistic studies by enabling the precise manipulation of signalling pathways and genetic modifications through small interfering RNA (siRNA) or CRISPR techniques [62,63]. The Langendorff perfused isolated heart model is commonly used in cardiovascular research to characterise the impact of global and regional ischaemia and reperfusion, and assess the efficacy of new cardioprotective therapies on small and large animal models [64]. However, there are some limitations in these modalities, such as a lack of appropriate disease models and systemic interactions, such as neurohumoral regulation and immune cell involvement, restricting their translational potential [65]. Even with these limitations, these methods remain a fundamental tool for screening cardioprotective agents and interpreting molecular mechanisms of IRI. Combining these models with co-culture systems or organ-on-a-chip technologies could boost their physiological relevance and narrow the gap between *ex vivo* and *in vivo* research [58].

### 4.2. In Vivo Models

Small animal models such as rats and mice are commonly used *in vivo* research of myocardial IRI due to their genetic manipulability, cost-effectiveness, and physiological similarities to human cardiac pathology. The most used mode of injury in *in vivo* models is left anterior descending (LAD) coronary artery ligation, which mimics the clinical setting of MI, followed by reperfusion [66]. In this model, transient occlusion of the LAD coronary artery, typically around 30–60 min, induces ischaemia, followed by reperfusion upon suture removal, activating common mechanisms underlying IRI, such as oxidative stress, calcium overload, mitochondrial dysfunction, and inflammatory responses, similar to human IRI [67].

LAD artery ligation allows for the investigation of cardioprotective strategies, including pharmacological interventions, gene modifications, and stem cell therapy [68]. Genetically modified animals, such as Spontaneously Hypertensive Rats (SHR) and diabetic rats and Atherosclerosis-prone apolipoprotein E-deficient (Apoe^−/−^) mice, provide helpful insight into molecular mechanisms in cardiovascular diseases [64,69,70]. However, challenges include high perioperative mortality, variability in infarct size, and difficulty in maintaining consistent reperfusion outcomes. Additionally, rodent hearts exhibit higher heart rates and different electrophysiological properties compared to humans, which may limit translational potential [58]. Further investigations utilising alternative large animal models, such as porcine and sheep, yield broader and more translatable insights due to their closer resemblance to the complexities of human cardiac physiology [71,72,73]. Nevertheless, unlike rodent models, particularly genetically modified or diseased mice, larger animal models are typically healthy, which limits their representation of pathological conditions.

Extrapolation of results obtained from animal models of ischaemic heart disease requires careful assessment of the advantages and drawbacks of each model. Small animals like mice and, to a lesser degree, rats provide transgenic disease models that could assist in monitoring changes in the diseased heart over their relatively short life span. Nevertheless, key physiological differences in heart rate, being faster in small mammals as well as basal metabolic rate, should be carefully considered in the interpretation and translation of the obtained data into clinically relevant study designs [74]. In this regard, despite being labour-intensive and costly, large animal models are, however, of great value to studies investigating cardiac ischaemia at structural and functional levels and in clinically relevant settings.

On the other hand, despite the widespread use of *in vivo* animal models in IRI research, significant translational challenges hinder the extrapolation of preclinical findings to human patients. One of the major limitations is that most experimental models lack the comorbidities commonly present in human IRI patients, such as diabetes, hypertension, and atherosclerosis, which have a significant impact on myocardial response to injury and treatment efficacy [68]. The variability in infarct size, extent and location, as well as healing and response to interventions between species, also complicates direct translation [75]. Furthermore, many preclinical therapies that show promise in animal models failed in clinical trials, emphasising gaps in mechanistic understanding and differences between species [76].

Despite these limitations, animal models remain indispensable for studying IRI pathophysiology and testing novel therapeutic approaches before advancing to clinical trials. Refinements such as echocardiographic monitoring and real-time molecular imaging continue to enhance the clinical relevance of these models and bridge the gap between bench and bedside.

### 4.3. Alternative Models and Approaches

Recent approaches in organoid technology and computational modelling offer novel advancements to studying IRI experimental models, addressing limitations in traditional *ex vivo* and *in vivo* models [77]. Cardiac organoids, derived from human-induced pluripotent stem cells (hiPSCs), replicate the three-dimensional architecture and cellular heterogeneity of human myocardium, allowing for more physiologically relevant studies of IRI [78]. These models facilitate the investigation of human-specific responses to IRI, overcoming species-related differences witnessed in rodent studies [79]. However, challenges eventually remain, including immature electrophysiological properties and the lack of vascularisation, limiting their ability to fully recapitulate complex I/R dynamics [58].

Complementing experimental approaches, computational models simulate IRI at cellular and tissue levels, integrating mathematical frameworks to predict molecular metabolism, infarct progression, and possible therapeutic effects [80]. These models enhance mechanistic understanding and enable high-throughput screening of cardioprotective strategies, reducing reliance on animal testing. However, computational models are limited by the need for extensive experimental validation, the challenge of accurately modelling heterogeneous patient-specific responses, and, most importantly, the high costs of experimental studies [81]. Integrating organoids and computational models with traditional approaches offers a promising direction for connecting the translational gap in IRI research, improving precision medicine strategies and accelerating therapeutic development in experimental models in IR research.

## 5. Current Therapeutic Strategies and Advances in IRI Research

### 5.1. Pharmacological Interventions

In recent years, advancements in the treatment of IRI have demonstrated promising improvements in clinical outcomes. This section aims to critically examine relevant therapeutic strategies concerning the underlying mechanisms and pathophysiological processes of IRI previously discussed in the review. As mentioned beforehand, oxidative stress is a central contributor to the pathogenesis of IRI, largely driven by excessive ROS generation during reperfusion [17]. This surge in ROS results in lipid peroxidation, protein modification, DNA damage, and eventually, cell death [7]. As an outcome, antioxidant therapy has emerged as a key pharmacological strategy aimed at controlling oxidative injury and improving myocardial recovery [27].

Agents such as N-acetylcysteine, allopurinol, and vitamin C have been explored for their ROS-scavenging properties. N-acetylcysteine, a precursor to glutathione, has shown efficacy in decreasing oxidative stress and myocardial damage in both preclinical and clinical stages [82,83]. Moreover, mitochondrial-targeted antioxidants like MitoQ have been developed to more precisely diminish mitochondrial ROS production, a main contributor to IRI development [84,85]. Other pharmacological interventions showing promising outcomes are ischaemic preconditioning (IPC) and postconditioning (IPOC), which have a powerful endogenous protective mechanism against IRI. This approach is characterised by brief episodes of ischaemia applied before or immediately after a prolonged ischaemic insult [21]. These strategies activate a cascade of molecular signals that mitigate cellular injury, reduce infarct size, and preserve cardiac function [86]. Pharmacological agents that mimic these protective effects, termed pharmacological pre- and postconditioning agents, have gained increased interest as potential therapeutic interventions, as they simulate the mechanical mechanism of action of IPC and IPOC [8]. IPC and IPOC mediate their effects primarily through activation of prosurvival kinases such as those in the Reperfusion Injury Salvage Kinase (RISK) and Survivor Activating Factor Enhancement (SAFE) pathways, which regulate mitochondrial function, inhibit apoptosis, and reduce oxidative stress [15]. IPC agents such as bradykinin, opioids, adenosine, and volatile anaesthetics like sevoflurane have demonstrated cardioprotective efficacy in preclinical models by targeting these pathways [23]. However, large clinical studies have failed to demonstrate similar outcomes to pre-clinical models. For example, the AMISTAD-II trial investigated the effect of adenosine as a preconditioning agent to induce cardioprotection, however, the patients did not report improved outcomes such as re-hospitalisation for heart failure [87]. This could be explained by variation in the time of administering adenosine in the early phase of ischaemia, where the myocardium is most salvageable [88]. Another clinical trial, DANAMI 3-iPost, investigating the effects of IPOC using balloon inflation within the reperfused coronary artery during PCI showed no significant difference in the end point of the study [89]. This could be attributed to multiple factors, including the variation of the duration of ischaemia in patients from onset until revascularisation [90]. Despite promising results in experimental models, the clinical translation of pharmacological interventions for IRI, including antioxidants and conditioning agents, has remained inconsistent. Factors such as patient age, comorbidities, timing of administration, and the complexity of oxidative and prosurvival signalling pathways contribute to the variability in therapeutic efficacy [84]. Consequently, current research is increasingly focused on optimising delivery methods, refining dosing protocols, and identifying patient populations most likely to respond to these treatments. These strategies aim to enhance translational success and reinforce the potential of pharmacological strategies as viable cardioprotective interventions in the management of IRI.

### 5.2. Molecular Targets and Gene Therapy

Targeting inflammation has emerged as a promising therapeutic approach in managing IRI, given the crucial role of the innate immune response in myocardial damage. Among key molecular targets is interleukin-1 (IL-1), a pro-inflammatory cytokine, which has gained significant attention [34]. IL-1 promotes leukocyte recruitment, endothelial activation, and the amplification of inflammatory cascades, contributing to cardiomyocyte injury during reperfusion [27]. Pharmacological inhibition using IL-1 blockers such as anakinra, a recombinant IL-1 receptor antagonist, has shown potential in reducing infarct size and improving cardiac function in preclinical models and early-phase clinical studies [34].

Beyond IL-1, gene therapies aiming to modulate inflammatory signalling pathways, such as NF-κB or NLRP3 inflammasome inhibition, are being investigated to provide more durable cardioprotection [31]. These molecular strategies underscore the shift towards precision medicine in IRI, targeting upstream drivers of inflammation to attenuate myocardial injury and improve post-ischaemic outcomes [27].

As the role of gene-editing technologies is developing rapidly in almost all medical fields, gene-editing technology in IRI, particularly CRISPR-Cas9, has introduced novel possibilities for targeting the molecular underpinnings of the injury. By enabling precise genetic modifications, CRISPR offers the potential to correct or suppress genes implicated in pathological responses to IRI, such as oxidative damage, inflammation, apoptosis, and mitochondrial dysfunction [91]. Recent preclinical studies have demonstrated the feasibility of using CRISPR to downregulate deleterious genes, including those involved in NLRP3 inflammasome activation and pro-apoptotic signalling, which are central to myocardial injury during reperfusion [92].

Moreover, CRISPR-based activation or repression systems (CRISPRa/i) allow for modulation of gene expression without altering the DNA sequence, making them attractive for reversible therapeutic strategies [91]. These approaches could be used to enhance cardioprotective genes, such as *SIRT1* or *PGC-1α*, which play roles in mitochondrial biogenesis and stress resistance [93]. Despite these promising developments, clinical translation remains limited due to concerns regarding off-target effects, delivery efficiency, and ethical considerations. Nevertheless, the integration of CRISPR with advanced delivery platforms and cardiac-targeted vectors holds promise for future gene therapies aimed at reducing IRI-related damage and improving myocardial recovery.

### 5.3. Stem Cell and Regenerative Therapies

Mesenchymal stem cells (MSCs) have emerged as a promising regenerative approach in the treatment of IRI due to their immunomodulatory, anti-inflammatory, and pro-angiogenic properties. Upon transplantation, MSCs secrete a broad range of bioactive molecules that mitigate myocardial damage, promote tissue repair, and enhance cardiac function [94]. However, increasing evidence suggests that the therapeutic efficacy of MSCs is largely mediated through their paracrine activity, specifically via extracellular vesicles (EVs) such as exosomes [95].

EVs derived from MSCs contain a complex cargo of proteins, lipids, mRNAs, and microRNAs that can influence key pathological processes in IRI, including oxidative stress, inflammation, and apoptosis [94]. For instance, MSC-derived exosomes enriched with cardioprotective microRNAs have been shown to diminish infarct size and preserve myocardial structure in preclinical models [95]. Additionally, EVs offer several advantages over cell-based therapies, such as lower immunogenicity, improved safety, and ease of storage and handling.

Even with the encouraging preclinical findings, significant challenges hinder the clinical translation of stem cell and regenerative therapies in IRI. Variability in stem cell sources, isolation techniques, and delivery routes contributes to inconsistent therapeutic outcomes [96]. Moreover, concerns about cell viability, immune rejection, tumourigenicity, and long-term safety limit widespread adoption [97]. While EVs derived from MSCs offer a cell-free alternative, issues such as standardised production, scalability, and precise cargo characterisation remain unresolved [95]. Regulatory hurdles and the lack of large-scale randomised clinical trials further complicate progress. Current research focuses on optimising EV engineering, dosing, and delivery strategies, making MSC-derived EVs a compelling candidate for future translational applications in cardiac IRI.

### 5.4. Biomarkers and Precision Medicine

Incorporating biomarkers and precision medicine into the management of IRI holds significant promise for improving therapeutic outcomes by modifying interventions to individual patient profiles. Identifying patient-specific vulnerabilities, such as comorbidities, inflammatory status, metabolic conditions, and genetic predispositions, can influence the selection and timing of cardioprotective strategies, thereby enhancing efficacy and minimising adverse effects [11]. Emerging biomarkers, including circulating microRNAs, cardiac troponins, cytokines, and ROS production indicators, provide real-time insights into myocardial injury and reparative responses [94]. These markers not only assist in early diagnosis and prognosis but also allow for the stratification of patients likely to benefit from targeted therapies [98]. Advances in omics technologies and machine learning are further enhancing the ability to interpret complex molecular data, identifying novel therapeutic targets and predicting individual treatment responses [99].

Adding to the point, the integration of omics technologies, such as genomics, transcriptomics, proteomics, and metabolomics, has significantly advanced the understanding of IRI, offering valuable insights into disease mechanisms and therapeutic targets [100]. These high-throughput approaches enable comprehensive profiling of molecular alterations during IRI, facilitating the discovery of novel biomarkers and eventually the development of personalised treatment strategies [97]. As an example, proteomic analysis has identified specific protein signatures associated with myocardial stress responses, while metabolomics has revealed metabolic shifts during reperfusion that may influence tissue recovery [100]. The porcine model of IRI was used to investigate the differential cardiac protein expression from early reperfusion and showed an increased expression of proteins involved in ribosomes, vesicle-mediated protein transport, in addition to collagen and extracellular matrix proteins in the ischaemic area. In contrast, mitochondrial and metabolism proteins were significantly diminished [101]. Metabolomic profiles after cardiac ischaemia demonstrate an impairment of fatty acid oxidation [102], as well as lactate and ketone body accumulation [103], suggesting a disruption to mitochondrial function, in addition to a decrease in essential and nonessential amino acids [104]. Interestingly, many of the metabolomic studies were notably carried out on patients who often other co-morbidities or metabolic diseases, reflecting a possible source of heterogeneity, which makes interpretation of metabolomics results quite challenging.

Moreover, omics data are increasingly being used to explore the efficacy of interventions such as remote ischaemic conditioning, allowing for the identification of patient subgroups most likely to benefit from such therapies [105]. Nevertheless, despite these improvements, significant translational limitations persist, as many research outcomes are derived from databases that may not fully replicate the complexity of human cardiovascular disease, particularly IHD, including critical factors such as age, comorbidities, and medication use [68].

By integrating biomarker profiling with clinical parameters, precision medicine approaches are redefining the landscape of IRI management, offering a shift from generalised treatment to patient-tailored care. Such strategies are vital for linking the translational gap and achieving more consistent outcomes in both experimental and clinical settings. In addition, the interpretation of omics data can be challenging due to inter-individual variability and the need for robust bioinformatic tools. Overcoming these limitations is essential for realising the full potential of omics-driven precision medicine as a therapeutic approach in IRI.

### 5.5. Future Perspectives: Nanotechnology and Biophysical Approaches and Biomaterial-Based Therapies

Nanocarriers have emerged as a promising strategy for enhancing targeted drug delivery in the treatment of IRI, offering improved therapeutic precision and reduced systemic toxicity [106]. These nanoscale delivery systems, such as liposomes, polymeric nanoparticles, and micelles, are engineered to transport cardioprotective agents directly to ischaemic myocardium, where they can exert their effects during the critical window of reperfusion [107]. Functionalisation of nanocarriers with targeting ligands or stimuli-responsive coatings enables site-specific drug release, increasing bioavailability and therapeutic efficacy [106]. For instance, antioxidant-loaded nanoparticles have shown success in scavenging ROS at the site of injury, while other platforms, on the other hand, have been designed to deliver anti-inflammatory or anti-apoptotic agents [84]. Moreover, nanocarriers can be tailored to bypass biological barriers and prolong circulation time, making them specifically useful in acute cardiac care [107]. Although still largely in preclinical phases, nanocarrier-based therapies represent a significant step towards precision medicine in the management of IRI.

Furthermore, therapeutic hypothermia and mechanical circulatory support (MCS) represent promising biophysical strategies for controlling IRI. Mild hypothermia (32–34.5 °C) has been shown to decrease myocardial oxygen demand, reduce inflammation, and limit infarct size by modulating cellular metabolism and apoptotic pathways [108]. Hence, when initiated early, it can preserve myocardial function during reperfusion and improve clinical outcomes afterwards.

In parallel, MCS devices such as intra-aortic balloon pumps and percutaneous ventricular assistance devices help to sustain coronary perfusion and unload the left ventricle during critical stages of ischaemia and reperfusion [109]. These devices not only support haemodynamics but also reduce myocardial workload and oxygen consumption, thus minimising further myocardial injury and unwanted complications [110]. Even with technical and logistical challenges in these therapeutic approaches, the integration of hypothermia and MCS into current treatment protocols will offer a synergistic and increasingly refined approach to cardioprotection in IRI.

In recent years, biomaterial-based therapies have emerged as promising adjuncts in MI treatment. Injectable hydrogels and biodegradable scaffolds are being developed to support tissue regeneration, modulate local inflammation, and deliver therapeutic agents directly to the infarcted myocardium [111]. Additionally, nanoparticle-based delivery systems have shown potential in preclinical models by enhancing targeted drug delivery and reducing systemic toxicity, with some platforms progressing to early-phase clinical trials [112,113].

### 5.6. Natural Compounds and Phytochemicals

Natural compounds and phytochemicals have garnered increasing interest as potential therapeutic agents in IRI due to their antioxidant, anti-inflammatory, and cardioprotective properties. Several compounds have shown promising results in preclinical models. Resveratrol, a polyphenol found in grapes and red wine, exerts protective effects through sirtuin 1 (SIRT1) activation, modulation of mitochondrial function, and attenuation of oxidative damage [114]. Curcumin, derived from turmeric, has demonstrated efficacy in reducing infarct size and suppressing inflammatory cytokines by inhibiting the NF-κB signalling pathway [115]. Berberine, an alkaloid from Berberis species, has been shown to protect against myocardial IRI by improving mitochondrial function and inhibiting cardiomyocyte apoptosis through AMPK activation [116]. Additionally, quercetin, a flavonoid, has been shown to reduce ROS formation and stabilise mitochondrial membranes [117]. These phytochemicals, while still under investigation, represent a promising class of adjunct therapies. However, challenges related to bioavailability, pharmacokinetics, and clinical translation remain to be addressed in future studies to harness their full therapeutic potential.

## 6. Limitations in Current Research and Future Interventions

### 6.1. Current Research Limitations

As previously discussed, the effective management of IRI continues to face significant limitations. These include translational barriers between experimental models and clinical application, inconsistencies and design limitations within clinical trials, and an incomplete understanding of the complex and multifactorial molecular mechanisms underlying IRI. Translating findings from experimental models of IRI to human clinical settings remains a major challenge due to fundamental interspecies differences in cardiac physiology, immune responses, and myocardial repair mechanisms [118]. Rodent models, while widely used, often fail to fully imitate the complexity of human cardiovascular disease, particularly in the presence of comorbidities such as diabetes, hypertension, or dyslipidaemia [68]. Furthermore, variability in experimental protocols, such as differences between *in vitro* H/R assays and *in vivo* MI models, can lead to inconsistent results and delay reproducibility [63]. Translational challenges to clinical practice include the heterogeneity of patient population in clinical trials, the presence of co-morbidities or medications that can affect ischaemic injury, variation in the duration of ischaemia and time of reperfusion, and the presence of adverse effects of newly developed therapies on other organs or systems [88,119]. These discrepancies extremely affect the reliability of preclinical data and eventually contribute to the high failure rate of therapeutic strategies in clinical trials.

Despite the great accomplishments in IRI research, significant gaps remain as an obstacle to understanding the key molecular pathways. Emerging regulated cell death mechanisms, such as ferroptosis and the complex interplay between autophagy and apoptosis, are not yet fully elucidated, restricting the development of targeted therapies in this field [7]. These processes appear to be highly context-dependent, influenced by timing, cellular environment, and disease state, making them often hard to study using conventional approaches [39]. Additionally, while omics technologies have generated vast datasets, the integration of multi-omics (genomics, proteomics, metabolomics) to construct a comprehensive view of IRI pathophysiology remains limited [100].

Hence, addressing these limitations in current research is essential for improving the predictive value of experimental models and advancing effective cardioprotective therapies for IRI. In addition, improved bioinformatic frameworks and systems biology approaches are needed to translate this complexity into actionable insights, thereby enhancing therapeutic precision and effectiveness of treatments.

### 6.2. Future Interventions

Future research approaches in IRI have demonstrated promising outcomes, highlighting the growing necessity to enhance key areas, including the refinement of experimental models, deeper exploration of underlying mechanisms, development of innovative therapeutic strategies, and the improvement of clinical translation pathways.

The development of humanised and translatable models, such as iPSC-derived cardiomyocytes and organ-on-chip systems, represents a significant advancement in IRI research [58]. These models more accurately recapitulate human cardiac physiology and disease conditions, enabling improved mechanistic insights and drug testing. Their integration into preclinical research holds promise for improving translational validity and closing the space between experimental findings and clinical outcomes [118]. Mechanistic studies in IRI are increasingly focusing on the role of epigenetic modifications and emerging regulated cell death pathways. Epigenetic alterations, including DNA methylation, histone modifications, and non-coding RNAs, have been involved in modulating gene expression during IRI and may offer novel therapeutic targets in the future [98]. In parallel, the exploration of alternative cell death pathways such as necroptosis and pyroptosis, both characterised by inflammation-driven cell lysis, has gained attention due to their important roles in myocardial damage during reperfusion [27]. Understanding the molecular regulation and context-specific activation of these pathways could reveal previously unrecognised mechanisms of injury and recovery. These emerging insights hold significant potential to refine therapeutic interventions and personalise treatment strategies in IRI based on molecular profiling.

Moreover, as artificial intelligence (AI) is the most recent revolution in the world, leveraging AI and machine learning in IRI research presents a promising avenue for therapeutic innovation [120]. These technologies enable the analysis of complex, high-dimensional datasets to accelerate drug discovery, predict treatment responses, and stratify patients based on molecular and clinical profiles [99]. AI-driven models can identify novel therapeutic targets and optimise clinical decision-making by integrating omics, imaging, and electronic health record data [121]. In the context of IRI, AI may improve the design of personalised interventions and enhance translational success by tailoring treatments to specific patient subgroups, defining their comorbidities and complex genetic profiles, hence ultimately contributing to more precise and effective cardioprotective strategies.

## 7. Conclusions

Ischaemia–reperfusion injury remains a significant contributor to myocardial damage in clinical contexts such as MI, cardiac surgery, and transplantation. This review has critically outlined the complex pathophysiology of IRI, highlighting key events including metabolic derangements, ATP depletion, ionic imbalance, acidosis, and the cascade of cellular damage that ensues during reperfusion. A deeper understanding of the cellular and molecular mechanisms underlying IRI, including oxidative stress, calcium overload, inflammation, mitochondrial dysfunction, and various forms of cell death, is essential due to their significant role in exacerbating myocardial injury and impairing recovery.

Current therapeutic approaches, including antioxidants, ischaemic conditioning techniques, anti-inflammatory agents, and emerging molecular therapies, have shown varying degrees of success in experimental models. However, translation to clinical practice remains hindered by several limitations, such as species-specific differences in experimental models, small and heterogeneous patient makeup in clinical studies, and an incomplete understanding of the multifactorial nature of IRI. Moreover, existing treatments are often constrained by issues of timing, dosage, and patient-specific variability.

Future directions in IRI research demonstrate promising avenues, particularly with the integration of advanced experimental preclinical models (e.g., iPSC-derived cardiomyocytes and organ-on-chip systems), novel mechanistic insights (e.g., epigenetic regulation and emerging cell death pathways), and therapeutic innovations such as gene editing, nanomedicine, and regenerative strategies. The application of AI and precision medicine will further improve the potential for personalised interventions.

In conclusion, cardiovascular diseases remain a leading cause of morbidity and mortality worldwide. Therefore, a deeper understanding of IRI pathophysiology, while addressing knowledge gaps on the complex and interconnected aspects of IRI such as mitochondrial dysfunction, calcium loading, oxidative stress and inflammatory response in pre-clinical models and patients, is essential to guide future therapeutic approaches. While considerable progress has been made in elucidating the mechanisms and therapeutic targets of IRI, substantial challenges remain in achieving effective clinical translation. Continued interdisciplinary research is essential to overcome current limitations, optimise therapeutic strategies, and ultimately reduce the burden of cardiac IRI on patients and healthcare systems.

## Figures and Tables

**Figure 1 biomedicines-13-02084-f001:**
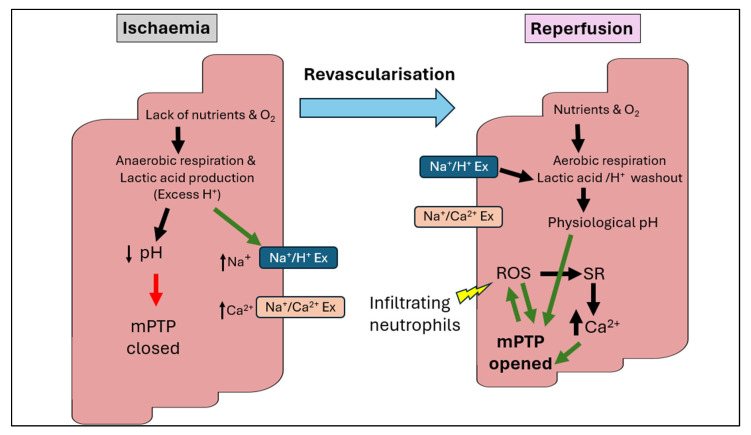
Overview of the pathophysiology of ischaemia–reperfusion injury. During ischaemia, excessive production of protons from anaerobic respiration induces the Na^+^/H^+^ exchanger to extrude H^+^ in exchange for Na^+^, consequently activating the Na^+^/Ca^2+^ exchanger to exchange Na^+^ with Ca^2+^. The acidic pH level initially prevents the opening of the mitochondrial permeability transition pore (mPTP) and cardiomyocyte contractures. During reperfusion, protons are washed out by further activation of the Na^+^/H^+^ exchanger. However, restoration of the physiological pH induces hypercontracture and releases the inhibition on mPTP opening, causing leakage of ROS and pro-apoptotic proteins into the cytoplasm. The calcium overload triggers mPTP opening and activation of cell death. The infiltrating neutrophils trigger further production of ROS, augmenting mPTP opening. Production of ROS activates sarcoplasmic reticulum (SR) to release more calcium, resulting in further damage to cellular targets such as lipids, proteins and DNA. Figure adapted from [21,22].

**Figure 2 biomedicines-13-02084-f002:**
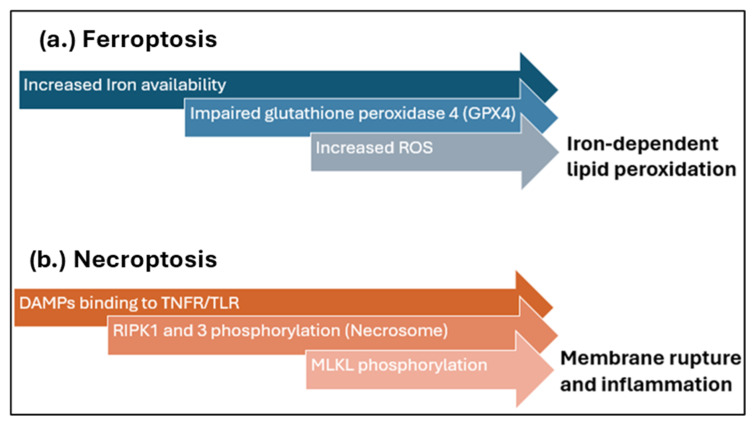
Key cellular events associated with cardiomyocyte death by (**a**) ferroptosis and (**b**) necroptosis following I/R. DAMPs, damage-associated molecular patterns; TNFR, tumour necrosis factor receptor; TLR, toll-like receptor: RIPK, receptor-interacting protein kinase; MLKL, mixed lineage kinase domain-like protein. Figure adapted from [41,42].

**Figure 3 biomedicines-13-02084-f003:**
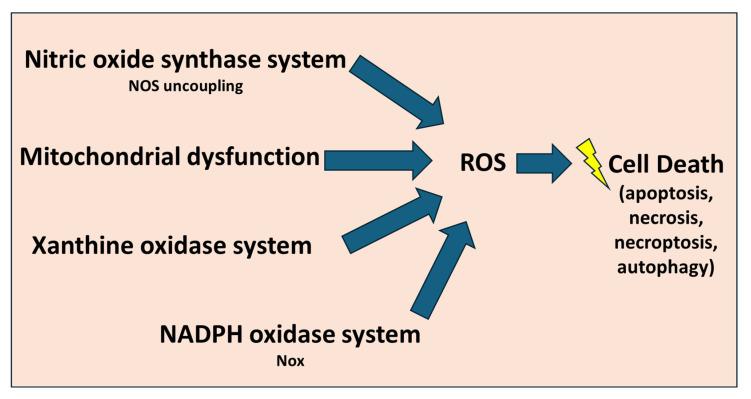
The main sources which produce ROS during IRI. These include the NADPH oxidases system (Nox), uncoupled nitric oxide synthase (eNOS) system, xanthine oxidase and the dysfunctional mitochondrial electron transport chain. The excessive generation of reactive oxygen species (ROS) during reperfusion depletes cellular antioxidant defense mechanisms, ultimately leading to cell death. Figure adapted from [17,31].

## Data Availability

No new data were created or analyzed in this study.

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
