# Peer review of "Cardiac Ischaemia–Reperfusion Injury: Pathophysiology, Therapeutic Targets and Future Interventions"

_biomedicines, 2025, doi:10.3390/biomedicines13092084_

Round 1

Reviewer 1 Report

Comments and Suggestions for Authors

The manuscript titled "Cardiac Ischaemia-Reperfusion Injury: Pathophysiology, Therapeutic Targets, and Future Interventions" is a comprehensive and well-structured review that effectively discusses the pathophysiology of cardiac ischaemia-reperfusion injury (IRI) and evaluates current and emerging therapeutic strategies. It reflects a commendable effort to integrate basic science with translational insights, emphasizing the clinical implications of IRI—an area of high relevance in contemporary cardiovascular research.

Major Concerns:

  1. While the manuscript outlines several therapeutic approaches, it tends to summarize rather than critically evaluate them. The limitations of clinical trials are only briefly mentioned. It is recommended that the authors include a comparative table summarizing failed versus successful clinical trials and provide a more in-depth analysis of translational challenges and reasons for failure.
  2. The manuscript leans heavily on previously published reviews and general references. While acceptable for broad overviews, the inclusion of more original experimental and clinical studies—especially recent ones—would significantly enhance the manuscript’s credibility. Particular emphasis should be placed on recent studies involving gene editing, mitochondrial-targeted antioxidants, and organoid models.
  3. Although ex vivo, in vivo, and in silico models are mentioned, their limitations are not critically discussed. The authors should include a comparative table that outlines the advantages and disadvantages of each model system, highlighting issues such as interspecies differences (e.g., rodent vs. human myocardium) and immunological variability.
  4. Some important claims, such as “MitoQ has been developed to reduce mitochondrial ROS,” are not adequately referenced. Direct citations to the original studies demonstrating such effects are essential.
  5. The manuscript contains an incorrect definition of “Ischaemia,” which could mislead readers. This must be corrected to maintain scientific accuracy.
  6. All figures included in the manuscript must be cited appropriately within the main text.
  7. The figure illustrating the pathophysiology of IRI needs substantial improvement. It should be revised to be self-explanatory and visually intuitive for readers.
  8. The authors are encouraged to include a section on natural compounds or phytochemicals that have demonstrated preclinical efficacy in IRI models and hold potential as future therapeutics.
  9. A dedicated section on biomaterial-based therapies for myocardial infarction (MI) and MI/IRI would enhance the scope of the review. This could include hydrogels, scaffolds, and nanoparticle-based interventions used in preclinical or early clinical studies.

Author Response

Comment 1:While the manuscript outlines several therapeutic approaches, it tends to summarize rather than critically evaluate them. The limitations of clinical trials are only briefly mentioned. It is recommended that the authors include a comparative table summarizing failed versus successful clinical trials and provide a more in-depth analysis of translational challenges and reasons for failure.

Response 1: Offering more discussion will direct readers to appreciate translational challenges rather than listing a case by case example in a table. Therefore, we decided to include in-text addition to the translational challenges of clinical trials, which are highlighted in yellow 

Comment 2: The manuscript leans heavily on previously published reviews and general references. While acceptable for broad overviews, the inclusion of more original experimental and clinical studies—especially recent ones—would significantly enhance the manuscript’s credibility. Particular emphasis should be placed on recent studies involving gene editing, mitochondrial-targeted antioxidants, and organoid models.

Response 2: Original studies are now added to the review

Comment 3: Although ex vivo, in vivo, and in silico models are mentioned, their limitations are not critically discussed. The authors should include a comparative table that outlines the advantages and disadvantages of each model system, highlighting issues such as interspecies differences (e.g., rodent vs. human myocardium) and immunological variability.

Response 3: We opted to include a separate paragraph to address this important point; however, due to the availability of numerous model systems, we decided to address this point generically, as it takes a separate review to offer a deeper scope into the different models.

Comment 4: Some important claims, such as “MitoQ has been developed to reduce mitochondrial ROS,” are not adequately referenced. Direct citations to the original studies demonstrating such effects are essential.

 Response 4: References to original studies were added in yellow

Comment 5: The manuscript contains an incorrect definition of “Ischaemia,” which could mislead readers. This must be corrected to maintain scientific accuracy.

Response 5: all ischaemia definitions have been edited and revised (in yellow)

Comment 6: All figures included in the manuscript must be cited appropriately within the main text.

Response 6: Figures are checked and cited in the main text where mentioned in text.

Comment 7: The figure illustrating the pathophysiology of IRI needs substantial improvement. It should be revised to be self-explanatory and visually intuitive for readers.

Response 7: The figure legend has been revised to aid in the visual understanding of the figure.

Comment 8: The authors are encouraged to include a section on natural compounds or phytochemicals that have demonstrated preclinical efficacy in IRI models and hold potential as future therapeutics.

Response 8: A section on natural compounds has been added.

Comment 9: A dedicated section on biomaterial-based therapies for myocardial infarction (MI) and MI/IRI would enhance the scope of the review. This could include hydrogels, scaffolds, and nanoparticle-based interventions used in preclinical or early clinical studies.

Response 9: A discussion on biomaterial-based therapies has been added.

Reviewer 2 Report

Comments and Suggestions for Authors

Major Comments:

  1. While the manuscript is rich in descriptive content, it lacks critical synthesis or discussion on contradictory findings across studies. Please consider incorporating a more evaluative tone by comparing study outcomes and highlighting inconsistencies or unresolved questions.
  2. The figure legends are overly descriptive and repetitive of main text. Additionally, Figures lack source citations. Streamline figure legends to focus on what each figure uniquely illustrates. Please also ensure figures are properly cited if adapted or re-drawn from other sources.
  3. Although translational gaps are acknowledged, the manuscript would benefit from a deeper discussion on the failure of past clinical trials. Please provide examples of failed IRI therapies in human trials and elaborate on what lessons have been learned to inform future research.
  4. The discussion on omics and precision medicine lacks concrete examples. Include specific biomarkers, genes, or omics-based studies that have led to actionable stratification in either preclinical or early clinical settings.
  5. Mitochondrial Dysfunction: This section is conceptually important but presented in a somewhat superficial manner. Consider expanding on the regulatory mechanisms of mitophagy and fission/fusion balance and how these are therapeutically modulated.

Minor Comments:

  1. The abstract is overly generic and lacks quantitative context. Please add specific numbers or outcomes from preclinical/clinical studies to emphasize impact.
  2. Certain phrases (e.g., “ROS production,” “oxidative stress,” “cardiomyocyte death”) are excessively repeated. Proofread for redundancy and streamline terminology.
  3. The stated aim is clear, but the manuscript does not clearly delineate what knowledge gaps exist. Add a brief paragraph summarizing key unanswered questions in the field.
  4. Please define all abbreviations at first mention in both main text and figures. Some (e.g., NETs, SAFE pathway) are not clearly explained.
  5. The conclusion lacks a strong take-home message. Please end with a concise 2–3 sentence paragraph summarizing the most promising therapeutic targets and the most urgent research needs.

Author Response

Comment 1: While the manuscript is rich in descriptive content, it lacks critical synthesis or discussion on contradictory findings across studies. Please consider incorporating a more evaluative tone by comparing study outcomes and highlighting inconsistencies or unresolved questions.

Response 1: We added evaluations to the different studies as highlighted in yellow

Comment 2: The figure legends are overly descriptive and repetitive of main text. Additionally, Figures lack source citations. Streamline figure legends to focus on what each figure uniquely illustrates. Please also ensure figures are properly cited if adapted or re-drawn from other sources.

Response 2: Figures sources are added, and legends are revised to avoid repetition with the main text.

Comment 3: Although translational gaps are acknowledged, the manuscript would benefit from a deeper discussion on the failure of past clinical trials. Please provide examples of failed IRI therapies in human trials and elaborate on what lessons have been learned to inform future research.

Response 3: Examples of failed clinical studies and explanations have now been added

Comment 4: The discussion on omics and precision medicine lacks concrete examples. Include specific biomarkers, genes, or omics-based studies that have led to actionable stratification in either preclinical or early clinical settings.

Response 4: more discussion on precision medicine and omics studies has now added (in yellow)

Comment 5: Mitochondrial Dysfunction: This section is conceptually important but presented in a somewhat superficial manner. Consider expanding on the regulatory mechanisms of mitophagy and fission/fusion balance and how these are therapeutically modulated.

Response 5: a paragraph has been added to illustrate this point (in yellow).

Comment 6: The abstract is overly generic and lacks quantitative context. Please add specific numbers or outcomes from preclinical/clinical studies to emphasize impact.

Response 6: Many thanks for your suggestion; however, it will be exceptionally challenging to address the specific impact of clinical studies in the abstract, as this can divert readers' attention from the wider scope of the review. Instead, we have added a quantitative figure to the impact of myocardial ischaemia in the abstract, emphasising the significant nature of the disease.

Comment 7: Certain phrases (e.g., “ROS production,” “oxidative stress,” “cardiomyocyte death”) are excessively repeated. Proofread for redundancy and streamline terminology.

Response 7: We used alternative terminologies where appropriate to address redundancy

Comment 8: The stated aim is clear, but the manuscript does not clearly delineate what knowledge gaps exist. Add a brief paragraph summarizing key unanswered questions in the field. L

Response 8: Revised the aim to address concrete examples of knowledge gaps

Comment 9: Please define all abbreviations at first mention in both main text and figures. Some (e.g., NETs, SAFE pathway) are not clearly explained.

Response 9: All abbreviations are checked and defined at first mention.

Comment 10: The conclusion lacks a strong take-home message. Please end with a concise 2–3 sentence paragraph summarizing the most promising therapeutic targets and the most urgent research needs.

Response 10: a takeaway message is now added to the conclusion on the direction of promising therapeutic targets and challenges ahead.

Reviewer 3 Report

Comments and Suggestions for Authors

I reviewed the manuscript very carefully, which is well written, presents a narrative sequence with adequate and solid scientific support, the figures are adequate and are mentioned and introduced in the manuscript in an appropriate manner, I did not detect any grammatical errors, in general I liked the article and it was pleasant and understandable to read, for this reason, I have no suggestions for the authors and I believe that it can be published in your prestigious journal as it is.

Author Response

Comment: I reviewed the manuscript very carefully, which is well written, presents a narrative sequence with adequate and solid scientific support, the figures are adequate and are mentioned and introduced in the manuscript in an appropriate manner, I did not detect any grammatical errors, in general I liked the article and it was pleasant and understandable to read, for this reason, I have no suggestions for the authors and I believe that it can be published in your prestigious journal as it is.

Response: Many thanks for your feedback and comments, much appreciated.

Round 2

Reviewer 2 Report

Comments and Suggestions for Authors

The comments raised have been addressed satisfactorily, and acceptable changes have been made in the revised text compared to the original manuscript.